# Lipidomic Profiling of Rice Bran after Green Solid–Liquid Extractions for the Development of Circular Economy Approaches

**DOI:** 10.3390/foods12020384

**Published:** 2023-01-13

**Authors:** Silvia Guazzotti, Cristina Pagliano, Francesco Dondero, Marcello Manfredi

**Affiliations:** 1Biological Mass Spectrometry Lab, Department of Translational Medicine (DiMeT), University of Piemonte Orientale, Via Solaroli 17, 28100 Novara, Italy; 2Center for Translational Research on Autoimmune & Allergic Diseases—CAAD, University of Piemonte Orientale, Corso Trieste 15/A, 28100 Novara, Italy; 3Department of Sciences and Technological Innovation, University of Piemonte Orientale, Viale T. Michel 11, 15121 Alessandria, Italy

**Keywords:** rice bran lipids, green solid–liquid extraction, ultrasound-assisted extraction, lipidomics, liquid chromatography-tandem mass spectrometry

## Abstract

Rice bran is a rather underutilized by-product of the rice industry that nowadays is far from being valorized. In this study, the lipidomic profile of bran of the Italian rice variety, Roma, has been evaluated through ultra performance liquid chromatography—tandem mass spectrometry. Crude lipid extracts were obtained from rice bran treated with different green solvents (1-butanol, ethanol and methyl tert-butyl ether/methanol mixture) in combination with an ultrasonic pre-treatment, and then compared with extracts obtained with standard solvents (chloroform/methanol mixture). Lipid yield, number and type of lipids and composition of prevalent lipid classes extracted were evaluated in order to provide an exhaustive lipid profile of the rice bran and to identify the most efficient green solvent for solid–liquid extractions. Twelve different lipid classes and a maximum of 276 lipids were identified. Ethanol and methyl tert-butyl ether/methanol solvents provided higher lipid extraction yields, the former being the most effective solvent for the extraction of triglycerides and N-acylethanolamines and the latter the most effective for the extraction of diglycerides, phospholipids and ceramides at 4 °C. Moreover, extraction with ethanol at 20 °C gave similar results as at 4 °C in terms of lipid yield and for most of the classes of lipids extracted. Taken together, our results indicate ethanol and methyl tert-butyl ether/methanol as excellent solvents for lipid extraction from rice bran, with the aim to further valorize this food by-product in the perspective of a circular economy.

## 1. Introduction

The two cultivated rice species, *Oryza sativa* L. (Asian rice) and *Oryza glaberrima* L. (African rice), provide one of the most widely harvested grains in the world (third only to corn and wheat) [1], and provide an important source of food for more than three billion people annually [2]. As the most populous country in the world, China consumes more rice than any other country, reaching about 154.9 million metric tons in 2021/2022, followed by India and Indonesia [3]. To produce the white rice kernels abundantly consumed around the world, whole grains undergo a milling process that generates by-products, such as husk, bran, germ and broken rice. Depending on the rice variety and the technique of cultivation employed, up to 40% of the total grain is lost as by-products during the milling process [4]. Rice bran (RB) is the most interesting by-product because although it accounts for about 10% of the grain’s weight, it contains around 65% of the nutrients of the whole grain [5], which has a positive impact on human and animal nutrition. Compared to wheat bran, RB contains a higher amount of fat matter, from 18 to 22%, which is a valuable resource once extracted [6]. However, due to its composition and the presence of an endogenous lipase, which hydrolyses fats into glycerol and fatty acids (FAs), RB is more susceptible to oxidation and thus not as suitable for baking [7]. As a result, a high proportion of RB is wasted every year or used as low-quality animal feed [8]. One of the most high-quality frequent uses of RB is the extraction of RB oil (RBO), which is primarily produced and consumed in Asian countries [9]. RBO is regarded as a “healthy oil” because of its well-balanced fatty acid content and the presence of beneficial components, such as γ-oryzanol and a mixture of antioxidant phytosterols with ferulic acid esters that can increase bile and neutral sterol excretion involved in cholesterol catabolism [10,11]. In addition, naturally occurring γ-oryzanol, along with vitamin E, exhibits pronounced free radical scavenging activity, which is helpful in protecting cells from oxidative stress [12], while phytosterols are responsible for RBO anti-inflammatory activity [13].

Over the years, some techniques have been proven effective in differentiating lipid compositions from various rice sources. Thin-layer chromatography was previously used to analyse lipids in RB [14,15], but due to the inability to identify and quantify lipid species, this technique has limited applicability in modern food analysis. In recent decades, gas chromatography- and liquid chromatography-mass spectrometry (GC-MS and LC-MS) have undergone exceptional development and use in lipidomics (especially for evaluating small molecules such as FAs) [16,17], and now ultra-high performance liquid chromatography-tandem mass spectrometry (UHPLC-MS/MS) has achieved a satisfactory ability to separate intact molecules with higher molecular weight and provide sufficient high-resolution chemical information suitable for untargeted lipidomic analyses [18]. According to recent studies, untargeted LC-MS based lipidomic approaches were successfully applied for the analysis of rice-lipid profiles and changes during storage [19,20] and during rice growth under hydric and heat stresses [21]. Such detailed lipidomic profiling is still missing for RB fraction. Lipid extraction from biological matrices, such as oil seeds and bran is generally carried out industrially through mechanical pressing or solvent extraction, with hexane as the most common solvent [6], and in the laboratory using organic solvents of fossil origin. Indeed, most lipidomic studies still rely on historical lipid extraction protocols with organic solvents first developed by Folch et al. [22] and Bligh & Dyer [23], which use a mixture of chloroform and methanol. Among these petroleum-derived solvents, hexane, which is employed industrially in large quantities, is dangerous for the environment and chloroform poses serious risks to human health since it is carcinogenic. In recent years, a new class of green solvents that are less harmful to human health, more environment-friendly and preferably bio-based emerged [24,25] and are today recommended or considered preferred in the solvent guidelines proposed by the GlaxoSmithKline and the Green Chemistry Institute-Pharmaceutical Roundtable and the CHEM21 European consortium [26,27]. As less dangerous alternatives, dichloromethane [28] and methyl tert-butyl ether replaced chloroform in a novel sample-extraction procedure for high-throughput lipidomics originally proposed in 2008 [29]. Considering more polar solvents such as alcohols, aqueous solutions of ethanol and 1-butanol have been used for lipid extraction from grains and bran, sometimes followed by lipidomic analyses. These alcohols are common bio-based solvents that can be derived from renewable feedstock since they are generated by bacterial or yeast fermentation of biomass [30,31]. As such, they meet all twelve principles of green chemistry described by Anastas and Kirchhoff [32], being environmentally friendly solvents derived from renewable sources. Regarding the potential use of physical methods to improve lipid yield, several studies performed on lipid extraction and ultrasonic pre-treatment of the matrix showed the formation of micro-bubbles. These bubbles, caused cell wall rupture and the release of intracellular components into the solvent [33] and affected the surface of the released oil droplets [34]. Another key parameter to consider during lipid extraction is the temperature, since a temperature decrease results in a reduced extraction yield, albeit alleviating lipid peroxidation [35].

The objective of the present study was to comprehensively evaluate the lipidomic profile of RB of the Italian rice variety, Roma (*O. sativa* L., subspecies Japonica), by analyzing through UHPLC-MS/MS the crude lipid extracts obtained from RB using different green solvents in combination with an ultrasonic pre-treatment. Untargeted lipidomic analyses have been used for the first time to compare the lipid extraction performance of the bio-based/potentially sourced renewably solvents methyl tert-butyl ether, methanol, ethanol and 1-butanol, with respect to the “gold standard method” based on chloroform and methanol [22] that has a worst environmental impact. The two main aims of the work were: (1) to search for a greener extraction method yielding a reasonable lipid yield, possibly exploitable also for further industrial purposes and (2) to provide an exhaustive lipid profiling of RB in terms of classes of lipids and individual molecules extractable with different solvents to valorise this food by-product in the perspective of a circular economy.

## 2. Materials and Methods

### 2.1. Reagents

The solvents used for lipid extraction and analysis, all of ACS grade, were methyl tert-butyl ether (Cas number 1634-04-4), methanol (Cas number 67-56-1), ethanol (Cas number 64-17-5), 1-butanol (Cas number 71-36-3) and chloroform (Cas number 67-66-3). All reagents used for UHPLC-MS/MS analyses were of LC-MS grade. Formic acid, ammonium formate and methyl tert-butyl ether were purchased from Merck (Darmstadt, Germany), water and acetonitrile from VWR (Milano, Italy) and methanol and 1-butanol from Scharlab (Barcelona, Spain).

### 2.2. Plant Material

Fresh-milled RB from the Italian rice variety, Roma (*O. sativa* L., subspecies Japonica), with a particle size < 0.5 mm was obtained from a local mill in North Italy. RB was stored at −80 °C and thawed at 4 °C just before the analyses.

### 2.3. Scanning Electron Microscopy Analysis

Scanning electron microscopy (SEM) images of dry RB samples were recorded using a Scanning Electron Microscope Leo 1450 MP at a voltage of 20 kV.

### 2.4. Lipid Extractions

Five different methods were used for solid–liquid extraction, based on different solvents or solvent mixtures and different temperatures, as detailed below:Chloroform/methanol (2:1, *v*/*v*) [19,22,23] at 4 °C (CH-ME);Methyl tert-butyl ether/methanol (3:1, *v*/*v*) [29,36] at 4 °C (MTBE-ME);Water-saturated 1-butanol [37] at 4 °C (WSBU);Ethanol (99%) [35] at 4 °C (ET);Ethanol (99%) [35] at 20 °C (ET20).

Briefly, for each experiment, 100 mg of RB were mixed with 2 mL of each solvent mixture (1:20 solid-to-solvent ratio (*w*/*v*)) and sonicated for 3 min using a sonicator (High Intensity Ultrasonic Liquid Processor, model VC 505 (Vibra Cell), equipped with a horn radiating face of 64 mm, with a frequency of 20 kHz, 40% amplitude and a pulsation of 10 s on/20 s off. All the extraction steps were performed at 4 °C, except for the extraction in ethanol at 20 °C (ET20). After sonication, the slurry mixtures were centrifuged at 3000× *g* for 10 min and the supernatants were collected. The solvent was evaporated from each extract using a centrifugal concentrator (Scanvac CoolSafe) at 20 °C and the extracts were stored at −80 °C until use for MS analyses. All the extraction experiments were performed in duplicate.

### 2.5. Determination of Lipid Yield

The yield of lipid extraction was calculated using the following formula:$$\mathrm{Lipid}\;\mathrm{yield}\;\left(\%\right)=\frac{\mathrm{extracted}\;\mathrm{lipids}\;\left(g\right)}{\mathrm{rice}\;\mathrm{bran}\;\left(g\right)}\;\times \;100$$

### 2.6. UHPLC-MS/MS Analysis and Lipidomics Data Processing

For the analysis of the lipidomic profile of RB, a UHPLC Vanquish system coupled with an Orbitrap Q-Exactive Plus (Thermo Scientific, Rodano, Italy) was used. Lipids were separated with a reverse-phase column (Hypersil Gold^TM^ 150 × 2.1 mm, particle size 1.9 µm) and kept at 45 °C at a flow rate of 0.26 mL/min. Mobile phase A consisted of acetonitrile/water 60:40 (*v*/*v*), mobile phase B of isopropanol/acetonitrile 90:10 (*v*/*v*) and both were modified with ammonium formate (10 mM) and 0.1% formic acid. The separation was conducted under the following gradient conditions: 0–2 min from 30 to 43% B, 2–2.1 min from 43 to 55% B, 2.1–12 min from 55 to 65% B, 12–18 min from 65 to 85% B and 18–20 min from 85 to 100% B; 100% B was kept for 5 min and then the column was allowed to re-equilibrate at 30% B for the other 5 min. The total run time was 30 min. All dried extracts were resuspended with pure methanol containing the internal standard 12-[(cyclohexylamino)carbonyl] amino]-dodecanoic acid (CUDA 12.5 ng/mL) as a quality control, and a sample volume of 3 μL was used for injection. The MS analysis was performed in positive ion mode with a voltage of 3.5 kV. Data were collected in a data-dependent (ddMS2) top-10 scan mode. The same procedural blank sample was examined to create an exclusion list for background ions and a quality control sample was examined right at the start of the analysis to ensure excellent consistency.

### 2.7. Statistical Analysis

All experiments were carried out at least in duplicate. The values reported are the means ± SD. The data were analyzed using one-way ANOVA and Tukey’s Honest Significance test. Significant differences were defined at *p* < 0.05, *p* < 0.01, *p* < 0.001 and *p* < 0.0001. Statistical analyses were performed using GraphPad Prism v.7 GraphPad Software and in www.graphrobot.com (accessed on 15 September 2022).

## 3. Results

### 3.1. SEM Morphology of Rice Bran

After the first removal of the inedible hull (i.e., the lemna and palea siliceous modified leaves) leading to the brown rice, abrasive milling removes the outer seed maternal tissues, producing the (milled) white rice and the by-products RB and polish. The bran, which is darker and more abundant than the polish, consists mostly of fragments of the more external pericarp, seed coat, nucellus, aleurone, and sub-aleurone layers and germ, with aleurone and germ components as the richest in lipids [38]. A picture of the raw bran of Roma rice used in this work is shown in Figure 1a. Observation of the sample using scanning electron microscope (SEM) revealed a heterogeneous particle size < 0.5 mm (Figure 1b) and the presence of aleurone layers (Figure 1c) rich in vacuolar inclusions, the phytin globoids (electron-dense spherical particles of 1–2 µm diameter) embedded in a protein matrix, surrounded by lipid droplets; a typical RB morphology [39].

### 3.2. Yield and Composition of Lipid Extracts

First, the different extraction methods were assessed for lipid yield (Figure 2). In all the extractions, lipid yields were between 11.7 and 19.3%, with ET (18.9%) and MTBE-ME (19.3%) being the most effective solvents, in accordance to previous results for lipids extraction from the same vegetable matrix [40,41]. Moreover, we observed that an increase in temperature from 4 to 20 °C (ET vs. ET20) and the addition of water to alcoholic solvent mixtures (WSBU) did not affect lipid yields, as shown in [35].

By analyzing mass spectrometry results obtained using 30 min chromatographic runs of the lipid extracts (Figure 3a), the lipidomic patterns of RB extracts were obtained (Figure 3b). During the first 16 min, polar lipids comprising the phospholipids (PLs) phosphatidylcholines (PCs), phosphatidylethanolamines (PEs) and phosphatidylinositols (PIs), the lysophospholipids (LPLs) lysophosphatidylcholines (LPCs) and lysophosphatidylethanolamines (LPEs), as well as few diglycerides (DGs), were eluted alongside N-acylethanolamine (NAEs) molecules and ceramides (Cers). During the subsequent 6 min, numerous neutral lipids, such as diglycerides (DGs) and triglycerides (TGs) were eluted, along with few molecules of hexosylceramides (HexCers) and sterols (STs).

Comparing the five different extracts, a maximum of 276 lipids were detected, belonging to 12 different lipid classes, with TGs, DGs and phospholipids such as PCs, PEs and PIs, accounting for the majority. Other less prevalent lipid classes were identified, such as Cers, HexCers, NAEs and STs. The diversity (i.e., number of classes) of the lipids extracted with the different methods and their relative abundances (expressed as percentage of lipids in each class relative to the total number of lipids extracted referred to each method) is shown in Figure 4. From this analysis, MTBE-ME extracted 276 species distributed in 11 classes, WSBU 261 species in 12 classes, ET 248 species in 10 classes, ET20 228 species in 10 classes, and CH-ME 194 species in 8 classes.

### 3.3. Identification of Phospholipids and Lysophospholipids

Phospholipids (PLs), also named glycerophospholipids, consist of a glycerol backbone with two hydrophobic fatty acid chains attached to sn-1 and sn-2 carbons and a polar head group attached to the sn-3 carbon through a phosphodiester linkage. Based on the type of polar head group at the sn-3 position of the glycerol backbone, PLs can be divided into several subclasses [42,43]. In this study, several phosphatidylcholines (PCs), phosphatidylethanolamines (PEs) and phosphatidylinositols (PIs) and only a single phosphoglycerol (PG) were detected (Appendix A). By comparing the five different extracts, we found a total of 41 PLs with MTBE-ME, 27 with WSBU, 24 with ET, 25 with ET20 and 21 with CH-ME (Appendix A).

PCs, PEs and PIs are known to be the most prevalent PLs in oil-rich RB, constituting 80% of the total PLs [44]. These PLs are the main constituents of the lecithins, extracted from vegetable foodstuffs and increasingly used as additives, emulsifiers and lubricants in pharmaceutical, cosmetics, paint and food industry [45]. The average composition of lecithin PLs extracted was in the order PCs > PEs > PIs, with proportions of 70–77%, 15–26%, and 4–8%, respectively, depending on the methods of extraction used (Figure 5a–c), in accordance with previous reports on vegetable analogues [46]. In all the lipid extracts, PCs were the most represented PLs (Figure 5a–c) and the MTBE-ME extract showed a significantly higher content of PCs with respect to the other methods (Figure 5a). Overall, MTBE-ME was the most efficient method to extract from RB all the three main PLs (Figure 5a–c), WSBU was valuable for PIs (Figure 5c); conversely, CH-ME did not perform in the extraction of this class of lipids and no PIs were found with it (Figure 5c).

Because of their great nutritional value, vegetable lecithins are now the most extensively utilized emulsifiers in the food and cosmetic industries. For instance, PCs have been used as a source of choline, a vital ingredient for the synthesis of acetylcholine, maintaining a healthy metabolism and aiding in the decrease in inflammatory processes and cholesterol levels [47]. Furthermore, esterified essential FAs can contribute to the nutritional value of PLs. In our research, oleic, linoleic, and palmitic acids were the predominant esterified FAs in the extracted PCs (Appendix A), in accordance to previous studies [48,49]. Our analysis revealed that all the extraction methods retrieved the majority of PC 36:3|18:1 18:2 and PC 36:2|18:1 18:1, with MTBE-ME showing an extraction efficiency of two or three times higher than the other methods.

Lysophospholipids (LPLs) are phospholipids in which one fatty acid chain is lacking (i.e., only one hydroxyl group of the glycerol backbone is acylated). In contrast to PLs, LPLs were only found in trace amounts in biological cell membranes [50]. In the lipid extracts from RB, some lysophosphatidylcholines (LPCs) and lysophosphatidylethanolamines (LPEs) were detected (Appendix A), with LPCs being mostly abundant in WSBU (Figure 5d) and LPEs in ET and ET20 (Figure 5e) extracts.

### 3.4. Identification of Ceramides and Hexosylceramides

Ceramides (Cers) and hexosylceramides (Hex-Cers), which are the basic components of sphingolipids and glycosphingolipids, play critical roles in several physiological processes and are fundamental structural components of the lipid bilayer of cell membranes [51]. In plants, Cers are composed of sphingoid bases (predominantly C18) esterified to long-chain FAs or very long-chain FAs (carbon chain > 20) [52]. In this work, 14 different Cers were detected with ceramide alpha-hydroxy fatty acid phytosphingosine and N-acylsphinganines (dihydroceramides) as the most abundant (Appendix A). MTBE-ME extracted the majority of Cers (10), resulting in the most efficient method for this class of lipids, followed by WSBU (7 Cers), ET (6 Cers), ET20 (4 Cers) and CH-ME (4 Cers), as shown in Figure 6.

HexCer (mono-hexosylceramides, monoglycosylceramides) belong to the group of cerebrosides within the sphingolipids. Their structure consists of a ceramide backbone linked to a neutral sugar molecule, ordinarily, glucose or galactose [53]. These molecules were detected only in MTBE-ME (5 HexCers) and WSBU (1 HexCer) extracts (Appendix A). Interestingly, HexCer 38:2;3O|18:2;2O/20:0;O (Appendix A), which has been demonstrated to increase epidermal HexCer metabolism in a human epidermal equivalent [54], compensating for epidermal loss of Cer, was identified only with the MTBE-ME method. In addition, other HexCers were identified (Appendix A).

### 3.5. Identification of Glycerides

In the outer layer of rice kernels, the glycerides, TGs and DGs, are the most abundant storage lipids and, here, DGs are also involved in the PLs biosynthesis pathway [55,56]. By comparing the five different extracts, a maximum of 152 TGs and 48 DGs were identified with ET, 144 TGs and 42 DGs with ET20, 159 TGs and 50 DGs with WSBU, 131 TGs and 28 DGs with CH-ME, and 145 TGs and 55 DGs with MTBE-ME (Appendix A). Considering the extraction of DGs (Figure 7a) and TGs (Figure 7b), all the methods showed similar efficiency with the exclusion of CH-ME, which was significantly lower.

In an industrial perspective, the abundant TGs and DGs extracted from RB might be recycled by creating a mono- and diglyceride combination with a free fatty acids (FFAs) level of up to 84% [57], which is often used as a food emulsifier under the code E471. Another industrial application might be their incorporation into biphasic gels systems [58], with superior ability to encapsulate both lipophilic and hydrophilic bioactive compounds.

Concerning the FA chains composition in the extracted DGs and TGs, our results are in agreement with the major literature [44]. Indeed, TGs positional isomers with unsaturated FAs in the sn-1 position were the most abundant components, while saturated chains, such as palmitic (16:0) and stearic (18:0) acids, were predominantly in sn-1 and sn-2 when present, as shown for extracts in ET (Figure 8a) and MTBE-ME (Figure 8b).

### 3.6. Identification of Other Minor Lipids

Other minor lipids, such as N-acylethanolamines (NAEs) and sterols (STs), were identified in rice-bran extracts (Appendix A). A NAE is a type of fatty acid amide formed when one of many types of acyl groups establishes a covalent bond with the nitrogen atom of an ethanolamine [59]. The precursor of NAE in plants, N-acylphosphatidylethanolamine (NAPE), was first discovered in wheat flour and various seeds from higher plants [60]. By comparing the five different extracts, seven NAEs were extracted with MTBE-ME, five with ET, four with CH-ME and WSBU and two with ET20 (Appendix A). According to our findings for NAEs, ET and CH-ME showed a significant superior extraction ability (Figure 9a). Besides NAEs, phytosterols are bioactive components naturally occurring in plants and mainly concentrated in rice germ and bran [61]. In our analysis, with the exception of CH-ME, all the other extraction methods detected three STs (Appendix A) and showed similar extraction efficiencies (Figure 9b). RB-derived oil contains more sterols than other plant-derived oils [62]. Sitosterol isoforms, in our MS pattern designated as ST 29:1;O with [M+NH_4_]^+^ adducts (Appendix A), were the predominant STs found in all the extracts, in accordance with other published evidence [63].

### 3.7. Principal Component Analysis (PCA)

Firstly, the normalized data, consisting of 3730 variables, were used for PCA modeling by autoscaling with Perseus (1.6.13.0 version) software. The study revealed that QC duplicates were tightly grouped, which was indicative of the instrument’s ability to produce consistent results for lipidomics. Chemical information from scores plot analysis widely supported the similarities of extraction with ET, ET20 and WSBU, which reflected the alcoholic nature of the solvents, in contrast to CH-ME and MTBE-ME clusters (Figure 10a). Figure 10b shows the loading plot, where, for instance, it is evident that LPE O-3:0, LPC 16:0 and TG 44:2 had large negative loading on PC2, whereas PC 36:2, PE 34:2 and PI 34:1 positively strongly influenced PC1.

### 3.8. Clustering Heat-Map

Next, we performed a clustering heat-map analysis of quantified lipids using Orange Data Mining (3.32 version) software. Data filtering of the top 50 modulated molecules was applied and normalization to interval [−1; +1] was used to preprocess the data. These lipids were selected as the most representative of all the extraction methods. The heat-map displayed one cluster for each extraction technique and highlighted the remarkable reproducibility of our findings (Figure 11). The clusterization confirmed our previous relative abundance results and evidenced that PLs were mainly represented in MTBE-ME, while LPEs (such as LPE O-3:0) and LPCs (such as LPC 16:0) were more abundant in WSBU and ET/ET20, respectively. In addition, Cers and HexCers predominated in MTBE-ME, with the exception of Cer 46:0, which was widely extracted in WSBU.

## 4. Discussion

Lipids derived from RB can be useful for several applications, including nutraceuticals and cosmetics, which are particularly devoted to the research of new molecules recovered from agricultural by-products [64,65]. The withdrawal of lipids from RB has traditionally involved the use of conventional solvents with a significant impact on the environment, with hexane as the solvent of choice. In the present research, four green extraction methods based on ethanol, 1-butanol and methyl tert-butyl ether/methanol mixture (ET, ET20, WSBU and MTBE-ME), according to the main sustainable solvents selection guidelines [64,66,67], were used for the extraction of RB lipids and compared to the gold-standard Folch method (CH-ME) [22]. The short chain alcohols methanol, ethanol and butanol are safer than the conventional solvents used for lipid extraction from RB and, currently, are considered valid alternative bio-based solvents for lipids extraction [68,69,70]. We did not consider primarily the cost of the selected green solvents; they are already bio-based or potentially sourced renewably, even though their production is currently low, and limited data are available on the actual costs, meaning that the potential costs need to be estimated. The extractions were performed at 4 °C to preserve the lipids and to perform a subsequent accurate lipidomic analysis. Moreover, to improve the extraction yield, sonication before the extraction was performed. ET was also used for extraction at a higher temperature of 20 °C (ET20).

Our data showed that ET and MTBE-ME extractions were the most efficient methods for total lipid recovery from RB, reporting about 19% of lipid yields, which is consistent with yields obtained with the use of conventional solvents [40]. Moreover, the total lipid yields were higher than those generally achieved with ethanol and isopropanol [35,71]. With regards to CH-ME, which is considered as a gold-standard procedure for total lipid extraction from tissues [72], the method gave the lowest lipid recovery (11.7%), in accordance with previous results obtained on rice bran [73]. Due to the lower efficiency in the CH-ME extraction, the results for LC-MS/MS areas, as well as the number of lipid molecules and their relative abundance within lipid families were negatively influenced. Regarding lipid classes’ relative percentages, TGs (72–86%), DGs (7–23%) and PCs (1–4%) were the most abundant lipids found in RB with all five tested methodologies, in accordance with previous results [43,49]. Overall, ET was the most effective approach for the extraction of TGs and NAEs, whereas MTBE-ME was the most effective strategy for obtaining DGs, PLs and Cers. Polar head groups and unsaturation degree in individual PLs classes are important factors contributing to lipid susceptibility to oil oxidation. For instance, PCs that were the most abundant type of PLs in all the extracts (with higher extraction rates in MTBE-ME) could be recycled for nutritional supplements as a source of choline and essential FAs, such as linoleic and α-linolenic acids [74]. In the cosmetic field, smaller and less saturated PLs chains can help the stabilization of oil-in-water emulsions, whereas long-chain PLs with unsaturated FAs bring to the formation of permeable liposomes [75]. In all the methods, the most abundant PCs were PC 36:3|18:1_18:2 and PC 36:2|18:1_18:1, which were already used to formulate emulsions and nanoliposomes [48,76,77], indicating a possible recycle of PLs extracted from RB in this field. HexCers and glucosylceramide (GlcCERs), with a single glucose head group, extracted more effectively with MTBE-ME, were previously extracted from rice and are currently commercially available [78]. A small amount of triene-type sphingoid base (sphingatrienine) has also been found in rice and maize [79]. For instance, the Glc-d18:2/h16:0 is already commercially available from RB and wheat germ in the form of skin moisturizer [80], whereas GlcCERs containing d18:2 acylated to -hydroxy FAs are the predominant species both in maize and rice [81]. While mice experiments highlighted that the inhibition of specific Cers genes could be a targeted approach for the treatment of obesity and type 2 diabetes mellitus [82], an oral administration of rice-derived GlcCers could compensate for the epidermal loss of Cer [54]. TGs are the most abundant lipids in rice bran [83]. Ethanol was particularly effective in extracting TGs, with similar efficiency either at 4 °C (ET) or at 20 °C (ET20). In previous studies, it was observed that higher extraction temperatures increased the yields of rice-bran oil [84,85]. However, temperatures higher than room temperature can speed up oil degradation and increase the amount of waxes and free fatty acids [86]. By comparing the lipid extracts obtained with ET and ET20, overall, the lipid yields and almost all the types of lipid molecules extracted at the two temperatures tested were not significantly different, making ET20 the most sustainable choice.

## 5. Conclusions

In this study, four green solid–liquid extraction and Folch methods were used to obtain total lipid extracts from rice bran. The samples were then analyzed in-depth using UHPLC-MS/MS to determine their lipidomic profile. In total, 12 lipid classes and a maximum of 276 different lipids were identified. ET and MTBE-ME were the most effective methods that gave higher lipid yields, comparable to the conventional extraction methods, with MTBE-ME extracting the highest amount of DGs, PLs and Cers, whereas ET extracting the highest amount of TGs and NAEs.

Overall, our results provide new insights into green-extraction methods combined to extensive lipidomic profiling of rice bran that can be exploited in a circular economy perspective of RB valorization for nutraceutical and cosmetic formulations, boosting bio-based solvents coupled to sonication for total lipid extraction from RB and minimizing residual wastes.

## Figures and Tables

**Figure 1 foods-12-00384-f001:**
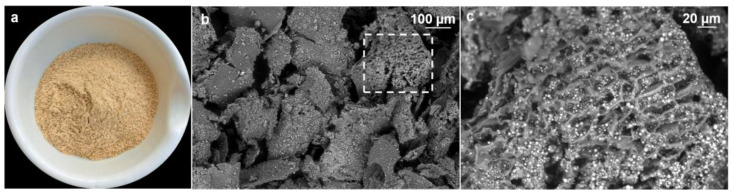
Picture of rice bran (**a**); SEM micrographs of rice bran at 100× (**b**) and 2500× (**c**). In (**c**) higher magnification of the boxed region in (**b**) shows aleurone layers with vacuolar inclusions.

**Figure 2 foods-12-00384-f002:**
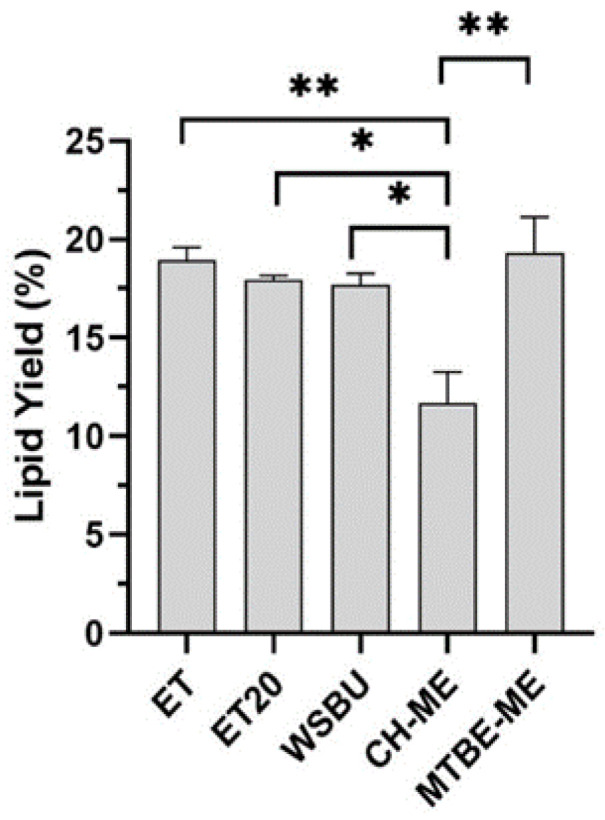
Yields of lipid extraction from rice bran obtained with different methods (mean ± SD of two replicates, * = *p*-value < 0.05, ** = *p*-value < 0.01).

**Figure 3 foods-12-00384-f003:**
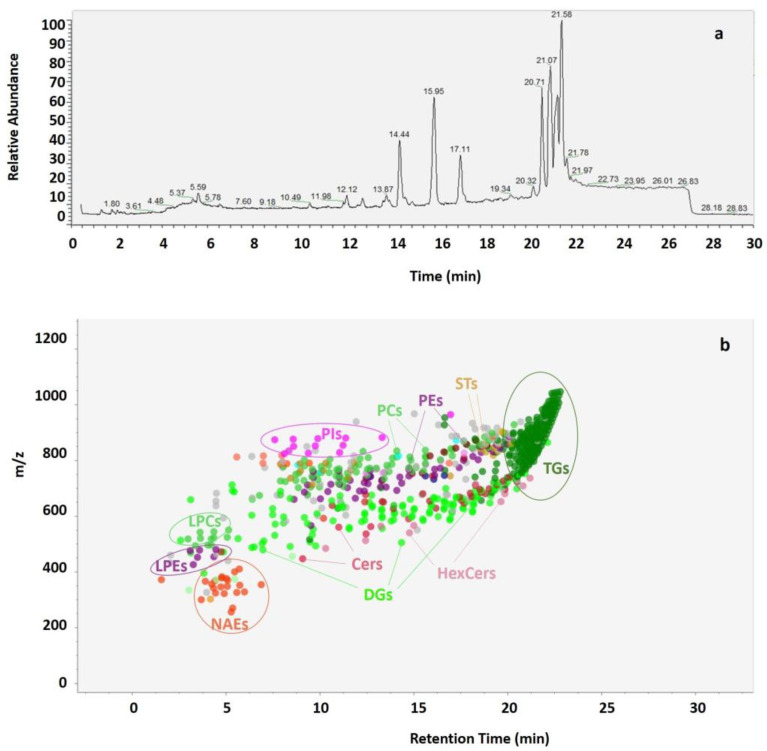
Example of UHPLC-Q-Extractive Orbitrap MS chromatogram acquired in positive ionization mode obtained from lipids extracted with MTBE-ME from a sample of rice bran (**a**). Spot viewer with m/z and retention time of different lipid classes (NAEs (orange), PEs and LPEs (purple), PCs and LPCs (green), PIs (fuchsia), Cers (red), HexCers (pink), DGs (light green), TGs (dark green) and STs (beige) (**b**).

**Figure 4 foods-12-00384-f004:**
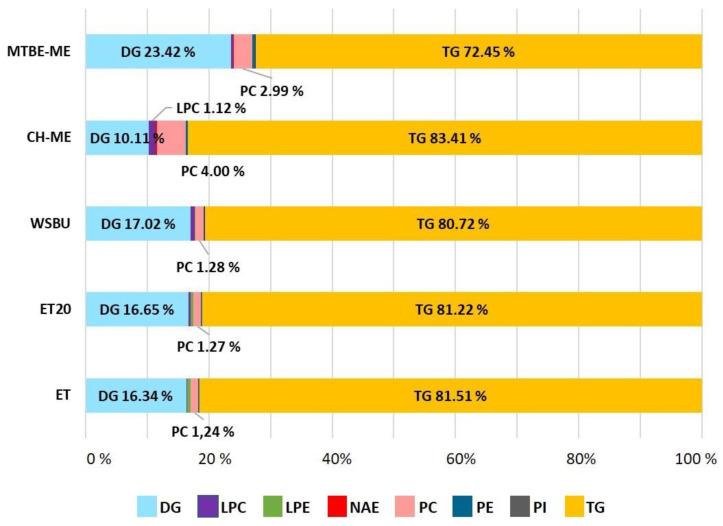
Content (%) of lipids distributed in different classes present in lipid extracts obtained with different extraction methods from rice bran (lipid classes with content < 1.00% are not shown).

**Figure 5 foods-12-00384-f005:**
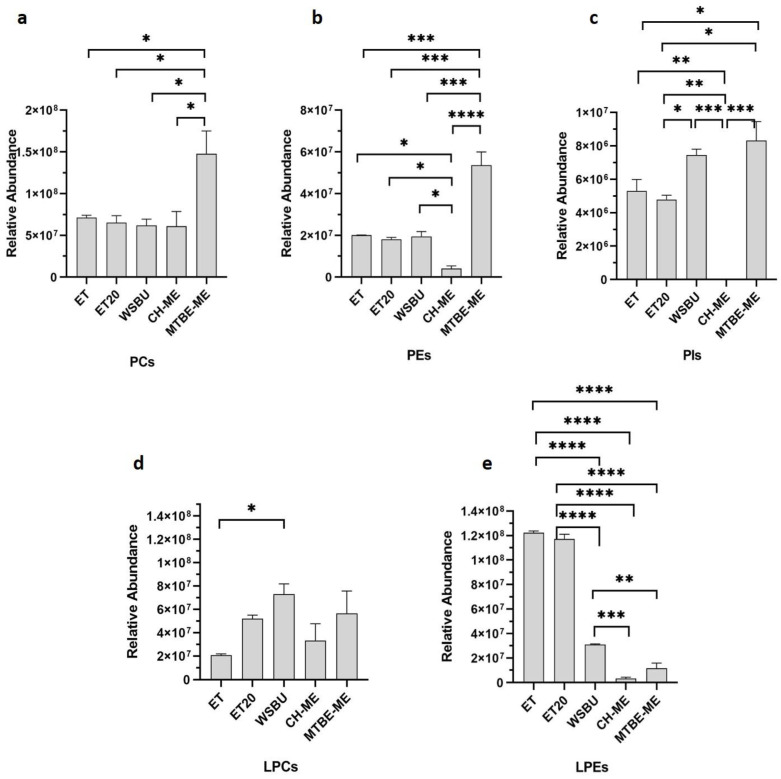
Relative abundance of lecithins in terms of PCs (**a**), PEs (**b**) and PIs (**c**) and of lysophospholipids in terms of LPCs (**d**) and LPEs (**e**) in different extraction methods (mean ± SD of two replicates, * = *p*-value < 0.05, ** = *p*-value < 0.01, *** = *p*-value < 0.001, **** = *p*-value < 0.0001).

**Figure 6 foods-12-00384-f006:**
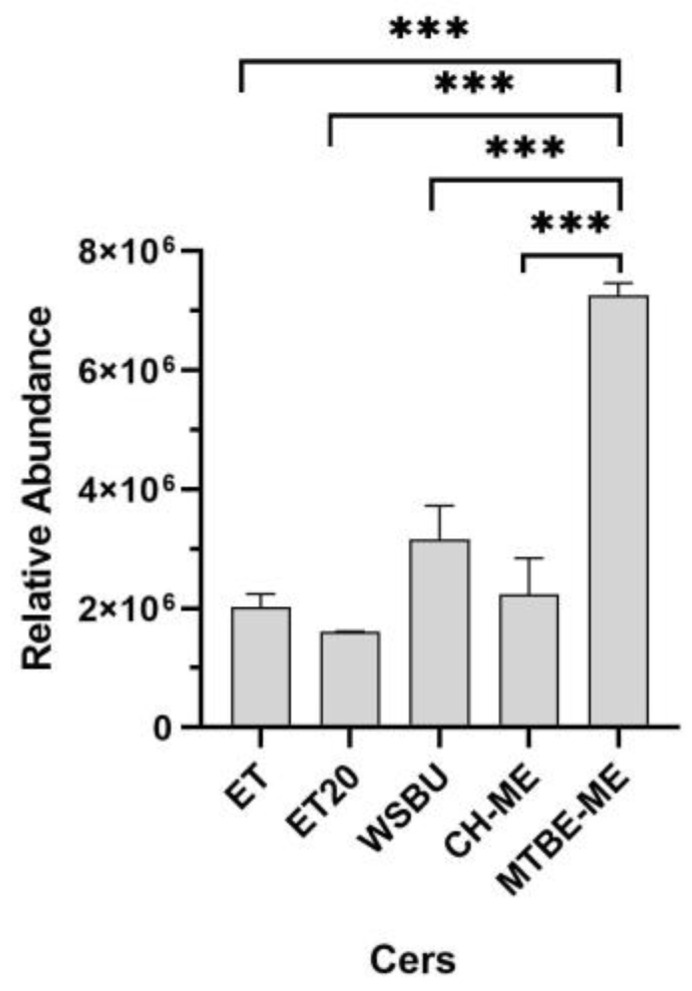
Relative abundance of ceramides in different extraction methods (mean ± SD of two replicates, *** = *p*-value < 0.001).

**Figure 7 foods-12-00384-f007:**
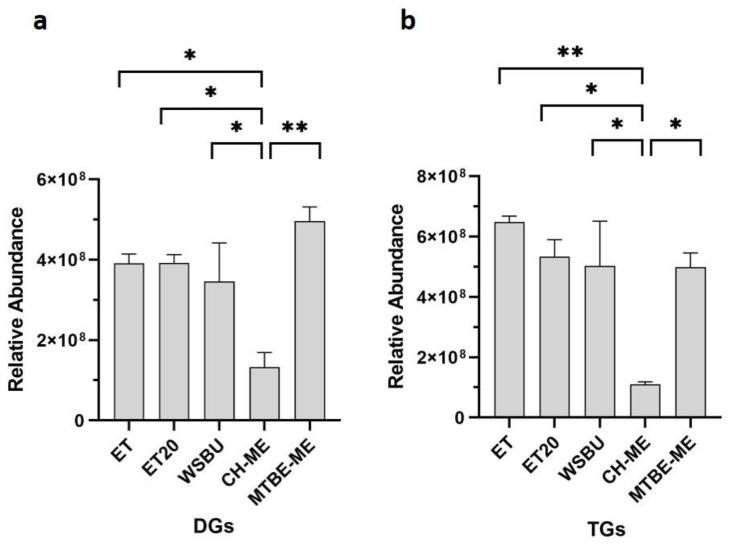
Relative abundance of glycerides in terms of DGs (**a**) and TGs (**b**) in different extraction methods (mean ± SD of two replicates, * = *p*-value < 0.05, ** = *p*-value < 0.01).

**Figure 8 foods-12-00384-f008:**
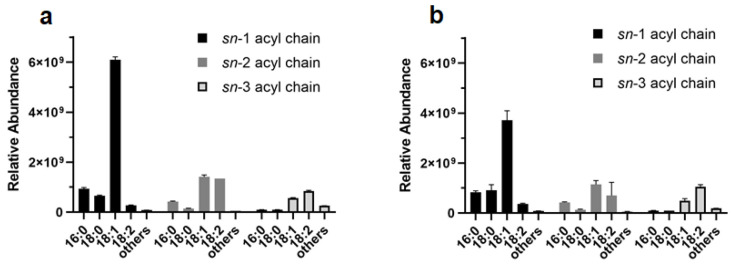
Relative abundance of acyl chains in DGs and TGs extracted with ET (**a**) and MTBE-ME (**b**).

**Figure 9 foods-12-00384-f009:**
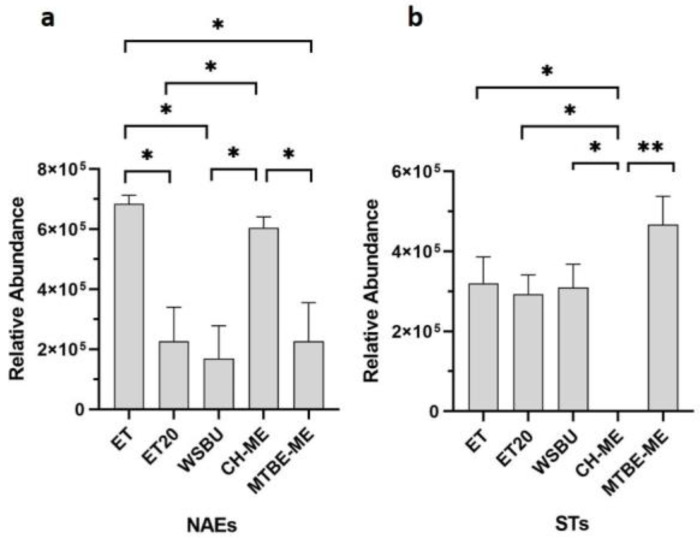
Relative abundance of NAEs (**a**) and STs (**b**) in different extraction methods (mean ± SD of two replicates, * = *p*-value < 0.05, ** = *p*-value < 0.01).

**Figure 10 foods-12-00384-f010:**
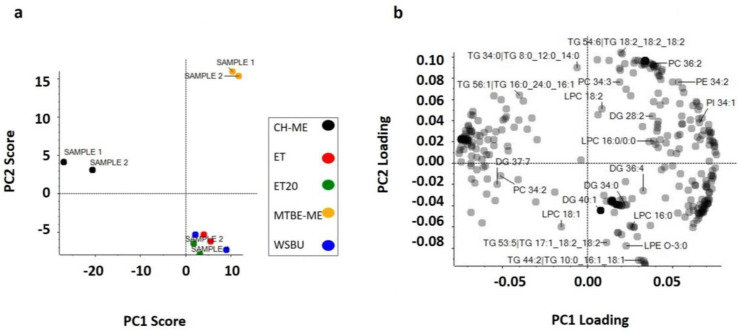
Principal component analysis (PC1 and PC2), with two duplicate scores for PC1 (47.5%) versus PC2 (23.8%) (**a**) and loading plot (**b**).

**Figure 11 foods-12-00384-f011:**
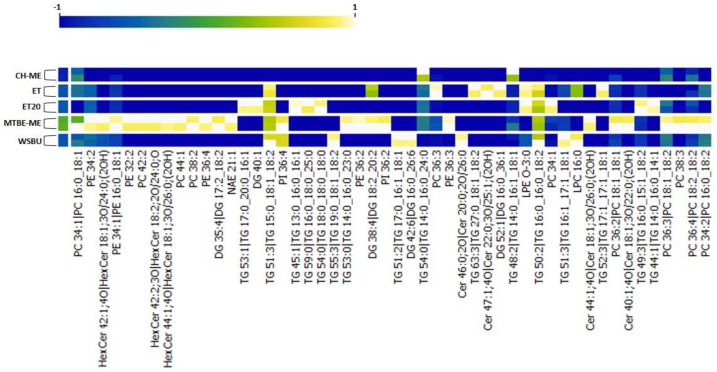
Heat-map of top 50 modulated lipids.

## Data Availability

Data are contained within the article and Appendix A.

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
