# Peer review of "Lipidomic Profiling of Rice Bran after Green Solid–Liquid Extractions for the Development of Circular Economy Approaches"

_foods, 2023, doi:10.3390/foods12020384_

Round 1
Reviewer 1 Report
In this paper, the authors proposed a greener extraction method of the rice bran in combination with ultrasound for the lipid identification using by UHPLC/MS/MS. In this context, Green solvents such us ethanol, t-butyl methyl ether, n-butanol and methanol were used. As result, ethanol and t-butyl methyl ether/ MeOH showed the higher lipid extraction index. Finally, some 276 lipids belonging to a twelve lipid groups were identified
The work is interesting as alternative to classic solvents. Maybe, the novelty is the use of Green solvents along with untargeted lipidomics of Italian rice bran. I have four questions about this work and the authors should justify it.
1. 2-MeTHF and CPME is considered as solvent alternative to dichloromethane and t-butylmethyl ether, respectively. They are considered a Green solution in extraction process. Why not use it?.
2. Ethyl lactate could replace to methanol. MeOH es considered toxic solvents and forbidden in food industry. What do authors think.
3. The use of n-heptane is GRAS through scientific procedures. Why not included it.
4. in relation a UAE. The frequency between 30-40 kHz for extraction is suggested. The authors used 750 Hz. Why
Author Response
See file attached

Reviewer 2 Report
1.The Keywords need to be revised to cover the main content of the manuscript.
2.The serial number of “Five different methods” Part 2.4 is changed to (1), (2)...
3.Figures need to be embellished for font and clarity.
4.Whether the authors considered the cost and toxicity of the solvents selected?
5.Some references need to be updated, and the format of the references should be revised according to the requirements of the journal.
Author Response
see file attached

Reviewer 3 Report
In my opinion, the manuscript entitled Lipidomic Profiling of Rice Bran after Green Solid-Liquid Extractions for the Development of Circular Economy Approaches by Guazzotti et al., is a good one. The introduction provides enough and a little too much information about rice bran, its chemical composition and a good study regarding the solvent used in general for green extractions, highlighting pros and cons. Materials and methods are enough described, results are good highlighted and compared with the current state of the art.
I have some comments and suggestions, as follows:
1. All supplementary tables should be added in pdf form with a description of the used abbreviations, so that readers could easier understand the samples codes. Results should be expressed with two decimals and followed by standard deviation (if they were made in duplicate, as authors mentioned).
2. line 403, extraction instead of extration
3. line 405, instead of alpha please use α
4. Line 420, effective instead of effetive
Author Response
see file attached
